# Robust reconstruction of cardiac T1 maps using RNNs

**Nicola Martini** [1]                                                                NICOLA.MARTINI@FTGM.IT
**Alessio Vatti** [1]                                                                 ALESSIO.VATTI@FTGM.IT
**Andrea Ripoli** [1]                                                                 RIPOLI@FTGM.IT
**Sara Salaris** [1]                                                                  SARA.SALARIS@FTGM.IT
**Gianmarco Santini** [1]                                                             GIANMARCO.SANTINI@FTGM.IT
**Gabriele Valvano** [1]                                                              GABRIELE.VALVANO@FTGM.IT
**Maria Filomena Santarelli** [2]                                                     SANTAREL@IFC.CNR.IT
**Dante Chiappino** [1]                                                               RADIO1@FTGM.IT
**Daniele Della Latta** [1]                                                           DELLALATTA@FTGM.IT
[1] *Deep Health Unit - Imaging Department - Fondazione Toscana G.Monasterio, Pisa, Italy*
[2] *Institute of Clinical Physiology (CNR), Pisa, Italy*

**Editors:** Under Review for MIDL 2019

## Abstract

Cardiac magnetic resonance parametric T1 maps are typically reconstructed using non-linear fitting. However this method has limitations due to the high computational cost and robustness. In this study, a recurrent neural network (RNN) is proposed for the robust and fast reconstruction of cardiac T1 maps.

**Keywords:** RNN, T1 mapping, non-linear fitting, MRI

## 1. Introduction

Cardiac T1 mapping is an established technique that permits both spatial visualization and quantification of myocardial tissue alterations of T1, focal or diffuse, in different cardiac pathologies (Moon et al., 2013). Several techniques have been proposed for cardiac T1 mapping (Kellman and Hansen, 2014), all based on three main steps: 1) perturbation of the longitudinal magnetization using either a inversion or saturation pulses; 2) single-shot acquisition of images that sample the recovery curve of the longitudinal relaxation at a fixed cardiac phase in breath-hold; 3) pixel-wise fitting with a proper T1-recovery mathematical model to generate the T1 map. However, in case of patients unable to hold the breath, respiratory motion may occur with subsequent errors in the estimation of T1 maps. A possible solution is to apply an image registration step before fitting, but this is a challenging task due to the large variations in image contrast at different inversion times (TIs) (Xue et al., 2012). In addition, when using magnitude-reconstructed images, the fitting procedure has to be performed multiple times to recover the correct signal polarity, thus increasing the computational cost (Nekolla et al., 1992). In this work we propose a Deep Learning technique based on recurrent networks (RNNs) as an alternative method for the reconstruction of parametric maps in cardiac T1 mapping.

## 2. Materials and Methods

### 2.1. Training data

We randomly generated synthetic MR signal curves according to recovery model of the modified Look-Locker inversion recovery (MOLLI) sequence (Messroghli et al., 2004):

$$y = \left| A - B \cdot e^{\frac{-t}{T1^*}} \right|$$

$$T1 = T1^* \cdot \left( \frac{B}{A} - 1 \right)$$

(1)

Model parameters A, B and T1* were chosen in the ranges observed on real data. To improve the robustness of the network, we added random perturbations to the data. In particular, each training batch is comprised of three different types of data equally balanced:

- Ideal data. Signal curves generated according to the recovery model.

- Data with additive noise. Noise was Gaussian distributed with zero-mean and standard deviation equal to 5% of the equilibrium longitudinal magnetization, i.e. parameter 5% of $A$.

- Data with outliers. For each curve two points chosen randomly were perturbed by signal multiplication with a factor sampled from a uniform distribution in the range [0.3, 1.5].

Furthermore, to account for cardiac arrhythmias during the MOLLI sequence acquisition we introduced for all curves a variability in the time axis corresponding to heart rate in the range [30, 120] bpm with random variations of the R-R interval (SD=200ms).

### 2.2. Network

The architecture chosen for the implementation is the Recurrent Neural Network (RNN), since it is the most appropriate network for time-series analysis. The proposed RNN takes as input the longitudinal magnetization recovery curves, sampled at different inversion times, and returns an estimation of the model parameters, which are used to compute the T1 value and hence the parametric T1 map. The RNN is composed by a Long Short-Term Memory (LSTM) layer with 1000 unit cells followed by a fully-connected layer with three output units corresponding to the model parameters of Eq. 1. The LSTM layer is fed with $2^{16}$ curves and is trained for 5000 epochs. The cost function that guided the training was implemented as the sum of three terms that account for the loss contribution of each model

parameter as following:

$$loss_A = A_{pred} - B_{true} \cdot e^{\frac{-t}{T1^*_{true}}} - (A_{true} - B_{true} \cdot e^{\frac{-t}{T1^*_{true}}}) = A_{pred} - A_{true}$$

$$loss_B = A_{true} - B_{pred} \cdot e^{\frac{-t}{T1^*_{true}}} - (A_{true} - B_{true} \cdot e^{\frac{-t}{T1^*_{true}}}) = e^{\frac{-t}{T1^*_{true}}} \cdot (B_{pred} - B_{true})$$

$$loss_{T1^*} = A_{true} - B_{true} \cdot e^{\frac{-t}{T1^*_{pred}}} - (A_{true} - B_{true} \cdot e^{\frac{-t}{T1^*_{true}}}) = B_{true} \cdot (e^{\frac{-t}{T1^*_{pred}}} - e^{\frac{-t}{T1^*_{true}}})$$

$$(2)$$

The T1 curves are eventually computed from parameters and compared to the reference curves through a Mean Absolute Error function (MAE).

## 2.3. Performance Evaluation

A test phase was performed to assess the performance of the proposed method in comparison to non-linear fitting with the Levenberg-Marquardt algorithm. The test evaluation cannot be performed on real data since the real values of the curves parameters are unknown. Therefore, to deal with missing ground-truth, a realistic synthetic MOLLI dataset was built using parametric maps A, B, T1* obtained by the fitting operation on N=40 acquired MOLLI datasets. The network was also tested on synthetic data simulating motion in the MOLLI acquisition. To this end, the following spatial transformations were applied to a random pair of raw MOLLI images: a rotation of 15°, a 1.5 cm translation on the x axis, and a 1.5 cm translation on the y axis. Images were processed both with conventional non-

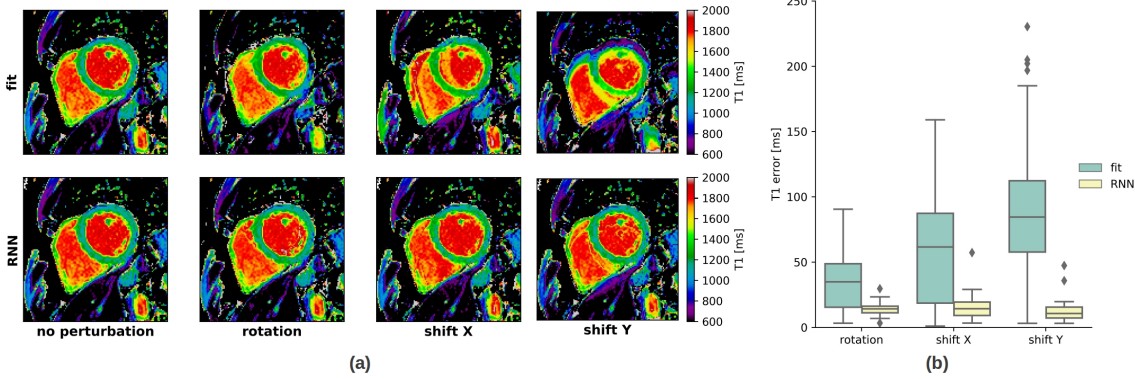

Figure 1: a) Reconstructed T1 maps with the conventional fitting and the proposed method for different types of synthetic MOLLI datasets. b) Mean T1 error in the myocardium.

linear fitting and the proposed RNN method. The errors committed by the two methods

are measured as the absolute difference between the estimated T1 and the ground truth T1 value.

## 3. Results and conclusions

As we can see from figure 1 a), the mathematical fitting is not able to estimate a correct T1 map in presence of motion. On the other hand, the RNN method provides a T1 map with higher matching to the ground-truth T1 map. This highlights the capability of the network to compensate for possible motion in the MOLLI images. Figure 1 b) shows the distribution of T1 error in the two method in the myocardium, under different spatial perturbations, demonstrating the higher robustness of the RNN with respect to the standard non-linear fitting operation.

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
