# OpenReview forum: "Robust reconstruction of cardiac T1 maps using RNNs"
_MIDL.io/2019/Conference/Abstract — MIDL Abstract 2019_

### Official Review · AnonReviewer2 · 2019-04-29
**Inconclusive results**

**Rating:** 2
**Confidence:** 2

**Review:**

The authors proposed to perform robust reconstruction of MR images with recurrent neural network (RNN) and validated the results on synthetically perturbed cardiac data. The results implied effectiveness of RNN on this task compared to conventional method while there might be some concerns about the results:

1) as the realistic datasets are not available, the author generated synthetic data with rotation and translation on x-  and y-axis. is there a way to know how close the synthetic data is to real data? Or in reality, can the data have a combination of those synthetic perturbations?

2) RNN is shown to be better than the conventional method, is there any recent work for robust reconstruction using neural networks and can be used as a baseline method?

---

### Official Review · AnonReviewer1 · 2019-04-30
**potentially promising**

**Rating:** 3
**Confidence:** 1

**Review:**

I am not familiar with this domain, so cannot really comment on the validity of the models for generating synthetic data. However, the overall approach of generating such synthetic data (with added noise, outliers, and arrhythmias) seems reasonable, and the proposed RNN approach seems to substantially outperform the alternate, nonlinear fitting approach.

---

### Decision · Program_Chairs · 2019-05-06
**Acceptance Decision**

Accept